# Sustainable Innovation in Organizations: A Look from Processes, Products, and Services

**Jhon Zartha [1],\*, Gina Orozco [1], Diana Barreto [2] and Diego García [2],\***

[1]  Faculty of Agroindustrial Engineering, School of Engineering, Pontificia Bolivarian University, Medellin 050031, Colombia; gina.orozco@upb.edu.co

[2]  Master's Program in Sustainability, Faculty of Agroindustrial Engineering, Universidad Pontificia Bolivariana, Medellin 050031, Colombia; diana.barreto@upb.edu.co

\*  Correspondence: jhon.zartha@upb.edu.co (J.Z.); diegoa.garcia@upb.edu.co (D.G.)

**Abstract:** Sustainability has been playing a major role on the world stage. As with everything, in the beginning, it was aligned with purely environmental contexts, but with the passing of the years, this concept has encompassed other aspects, such as the way of producing products and services. In this article, to obtain information on the implementation of sustainable innovation, a search equation was formulated in the SCOPUS database, focusing on the period from 2018 to 2023, with a total of 62 articles that are related to the topic addressed. Once the search of the published articles was carried out, seven categories were formed to classify the research topic of each of the publications with Bibliometrix software version 4.1.1 and the PRISMA method, according to the main objective and methodology used in each one. In this sense, 44% of the articles published are related to open innovation, and 26% of the articles are focused on sustainable business models. At the end of the analysis of the implementation of innovation according to the articles consulted, it was possible to establish the fact that sustainable innovation gave rise to some important work in the mitigation of adverse impacts.

**Keywords:** sustainability; innovation; eco-innovation; sustainable innovation; processes; products

## 1. Introduction

Sustainable innovation in organizations has been implemented in processes, products, and services worldwide and, in this sense, it is important to highlight that some research has addressed the application of sustainability from a circular economy approach in relation to these products and services [1].

Therefore, it is important to highlight that innovation at the organizational level can occur from the development of products and/or services, as well as in the modification of both administrative and operational processes, an aspect that has gained importance since it allows us to obtain effective results that have a positive impact on organizations. Therefore, the problem of determining how worldwide organizations manage sustainable innovation in their processes arises. In addition to the above, existing tools that address the issue of sustainable innovation, such as the Oslo Manual [2], do not do so in a profound manner and do not involve all the variables that integrate organizational strategies for sustainable innovation.

This is how most authors highlight the fact that sustainable innovation has allowed us to achieve a significant decrease in environmental impacts and sustainability in organizations; the authors of [3] state that there is a lack of marketing by companies to reduce the lack of consumer information on products and increase consumer interest in eco-innovation.

Concerning the above, it is important to delimit the causes that do not allow the implementation of sustainable innovation at the organizational level, within which we can highlight the lack of resources, deficient management, and the lack of will or leadership of

the top management in transforming the different types of knowledge and achieving its accessibility to all employees [4].

The purpose of this article is to identify the factors that have or have not allowed us to implement processes, product development, and services within the context of sustainable innovation, which will allow us to solve the following question: how has the process of implementing sustainable innovation at the organizational level in processes, products, and services been? This will allow us to analyze the different criteria that have been established at the organizational level to promote these practices, allowing different companies, regardless of their size, to contribute to development and sustainability. The above is based on business models, types of industry, and the way of implementing sustainable innovation in processes, products, and services. In this sense, articles were selected that present how to implement sustainable innovation worldwide in different organizational interactions.

This paper is divided into four sections as follows: The first presents a theoretical framework, which focuses on innovation, approaches to sustainable innovation, and the relationship between business and sustainability; the second section focuses on the methodology used through a search in the SCOPUS database with Boolean operators, performing an in-depth analysis on the selected documents. Once the search was carried out in the said database, the PRISMA method was applied to exclude articles that were not related to the research topic for any reason. Subsequently, a software called Bibliometrix was used to relate the variables used or identified in each study; finally, the third section presents the results obtained and a discussion highlighting the categories related to sustainable innovation and some barriers and drivers of sustainable innovation. The conclusions of the study are detailed at the end.

## 2. Literature Review on Sustainable Innovation

At present, the definition of sustainable innovation does not have a totally clear conceptual basis. However, it should be considered that it is one of the most studied areas and that a strategic component is also contemplated within it.

About 150 years ago, Marx and Engels made a comparison between innovation and a two-edged sword, making the allegory that it can be used to create or destroy. In the 20th century, it was posited that organizations would find a new opportunity in innovation to meet the needs of their customers [5].

Schumpeter was the first to define the concept of innovation in his three major works, as follows: *Theory of Economic Development*, *Capitalism, Socialism and Democracy*, and the *History of Economic Analysis*, where it is described as the constant search for benefits that can be received by the first person or company that introduces it [5]. Based on the above, it can be said that innovation facilitates market dominance, promotes investment in technology, and improves conditions so that the entry of new competitors does not affect, or at least is minimal to, the environment.

Sustainable innovation in organizational processes uses environmental, social, and economic approaches in its implementation so that a company dedicated to the development of products or inputs can achieve acceptance among different stakeholders. This is reinforced by the conceptualization discussed in [6], which indicates that innovation can be defined as the result of a set of activities that use knowledge, skills, and/or technologies to satisfy individual or social needs.

Currently, the vision of sustainable innovation has a more general scope because it involves strategic aspects that are framed in the planning of organizations, whether to produce products or to provide services. In the same way, there is a direct relationship between the use of technology and information, which promotes the conscious use of these resources to positively impact society and its environment by generating beneficial financial results.

Therefore, knowledge analysis and transfer are effective precursors of innovation, since knowledge and learning are considered to contribute to better production systems. [7],

talks about knowledge production and how the way of producing it has changed; in fact, he points out the emergence of a mode of production that implies, in principle, a difference from the traditional way of producing knowledge. Therefore, as innovation has been incorporated positively at all levels, its evolution has been inherent to the sustainable processes of the company.

Dosi, cited by [5], is oriented to technological processes, which should be focused on guiding innovators' processes that are aligned with the needs of the sectors where they are developed. On the other hand, Grant, cited by [5], maintains that the most effective technical innovations are born from the advantages of the integration of human and material resources since they can mainly determine their innovative performance.

Innovation is linked to creativity which according to Rodriguez Valencia cited by [8] is the ability to consistently generate different and valuable results. The human being puts this creativity at the service of accelerated development in organizations that impact the strategy of these companies, generating dynamics that range from the appropriation of new processes to generational change and the appropriation of knowledge.

- **Implementation of sustainable development worldwide**

The debate surrounding the concept of sustainable development is wide-ranging. Thinking about sustainable development is a subject that requires taking into account economic, social, and environmental aspects [9]. Given the above, it is necessary to generate a synergy that brings together innovation and sustainable entrepreneurship, to promote economic growth, social protection, and the preservation of natural capital.

This also calls for the business sector to establish in its strategies the use of environmentally compatible technologies so that activities rooted in environmental management are immersed in the organizational strategy as part of innovation. Under this analysis, the concept is evolving with others such as social responsibility, business sustainability, and corporate social responsibility, among others. In the business scenario, sustainability and innovation are contemplated as strategies for social, environmental, and economic growth in the countries where they operate contributing to their development. The above is supported by Tolotti cited by [10], who mentions that in times of crisis such as the recent COVID-19 pandemic, the use of strategies oriented to collective innovation can be the ideal scenario to face difficulties and thus maintain a desired quality of life.

The advancement of this relationship between innovation and sustainable development has brought about the development of several disciplines, which include today's nanotechnology, emerging technologies aligned to corporate governance and a legal framework that also involves society and the community in general and values this incorporation within the strategies that organizations undertake to remain sustainable over time. According to [10], sustainable innovation is a shared responsibility, which implies the expansion of governance mechanisms, with an inclusive approach that respects and values the environment, and social and economic dimensions.

- **Companies and sustainability**

Every day, more and more companies are working on innovative attributes in their processes or product manufacturing, all with the expected result of quality and on this occasion, sustainability. According to [11], the challenges faced by organizations to make the internal ecosystem sustainable, in which people's motivations and capabilities are articulated to innovate, create value, and nurture sustainable competitive advantages, impact sustainability.

Organizational innovation is based on a series of premises that characterize these behaviors, which can be seen in a global scenario.

Innovation in product research with a social, environmental, and ecological focus, for example, incorporating environmentally compatible raw materials.

Analysis of the critical aspects that have an impact on the technical–environmental evaluation of innovation processes based on the application of cleaner technologies and suppliers with sustainable criteria.

Participation of stakeholders involved in the different innovation processes, including the public and private sectors, for example, through accountability.

Regulations to minimize threats and maximize opportunities to face adverse situations in innovation processes stipulate a risk analysis in this respect.

Information and feedback mechanisms as a fundamental part of the innovation process.

The above indicates that, when talking about business and sustainability, more than a punctual knowledge of the subject is required; today, it is a field that demands multiple expertise that can form interdisciplinary teams that can enrich the process of evolution and implementation of sustainable innovation at the organizational level. For a long time, companies saw the problems of their people closer to human resources management and further away from the long-term sustainability challenges of the business [11], where the organizational solutions lie.

With the above, the field of research on how innovation influences organizations, their processes, and products has a very broad spectrum that needs to be explored because of the knowledge gaps in its implementation which could have a maximum impact at the strategic level. Ref. [12] argues that investment in research and development (R&D&I) improves the environmental perspective and innovation, which influences the proper functioning of organizations, resulting in their competitive capacity, quality of employment, and overall sustainability.

The methodology used in structuring this article consists of a descriptive documentary-type qualitative research according to [13], who highlights such approach as follows:

"One who uses data collection and analysis to refine research questions or reveal new questions in the process of interpretation."

Regarding the phases of the research, it was conducted as follows, taking into account authors such as [13,14].

We mainly selected articles that were published between 2018 and 2023 that were related to the analysis of sustainable innovation in both process, products, and strategy, using the PRISMA methodology, which can be seen in Figure 1.

### 2.1. Generation of Information

It consists of spatial and population delimitation, which, for the case study, will focus on a global level in any type of company. In this sense, a search equation will be applied to databases of indexed journals related to the subject. In this sense, a search equation will be applied to databases of indexed journals related to the subject.

The equation was created including the following variables:

AND: The database search engine will be instructed to present articles only with the terms indicated followed by "AND".
OR: The database search engine will be instructed to present articles with any of the terms described in the phrase
PUBYEAR: This will help to purge articles published after 2018—it should be before 2018. LIMIT TO
": The database search engine will be instructed to present articles only with the terms indicated within the quotation marks. For example: Sustainability, Sustainability, etc.

It is important to highlight that in the databases of indexed journals such as Scopus, the following search term was used, and a total sample of 62 English articles related to the focus of the research were obtained:

"sustainable innovation" AND "processes" AND "goods" AND "services" AND "products" AND "industries" AND "enterprises" AND PUBYEAR > 2018 AND ( LIMIT-TO ( OA,"all" ) ) AND ( LIMIT-TO ( PUBSTAGE,"final" ) ) AND ( LIMIT-TO ( DOCTYPE, "ar" ) ) AND ( LIMIT-TO ( PUBYEAR,2019) OR LIMIT-TO ( PUBYEAR,2020) OR LIMIT-TO ( PUBYEAR,2021) OR LIMIT-TO ( PUBYEAR,2022) OR LIMIT-TO ( PUBYEAR,2023) ) AND ( LIMIT-TO ( EXACTKEYWORD,"Sustainable" ) OR LIMIT-TO ( EXACTKEYWORD,"Sustainable Development" ) OR LIMIT-TO ( EXACTKEYWORD,"Innovation" ) ).

Once the equation in the SCOPUS database was used, we proceeded to identify the relationships of the variables through the Bibliometrix software, among which we can mention annual scientific publication, scientific production by country, main information, and word cloud, among others.

**Identification of studies via databases and registers**

**Identification**

Records identified from: Scopus.
Databases (n = 1)
Registers (n = 498)

Records removed *before screening*:
Duplicate records removed (n = 458)
Records marked as ineligible by automation tools (n = 0)
Records removed for other reasons (n = 336)

**Screening**

Records screened.
(n = 122)

Records excluded.
(n = 60)

Reports assessed for eligibility.
(n =62)

Reports excluded:
Not related to search terms (n = 40)
Different approaches to the study (n = 20)

**Included**

Studies included in review.
(n = 62)
Reports of included studies
(n = 62)

**Figure 1.** Identification of studies via databases and registers [15].

### 2.2. Consolidation of Information

The information collected in the previous phase is to be consolidated utilizing information recording instruments. In Table 1, the tasks of the activity associated with that phase will be explained.

**Table 1.** Approach in the information consolidation phase.

| | |
|---|---|
| Activity | We constructed the respective analytical summaries of information (ASI) for each of the identified articles, obtaining information related to the objectives, methodology, and conclusions (13). |
| | The information from the RAI was consolidated into an Excel matrix to allow the identification of the most relevant components of sustainable innovation in the organizations in processes, products, and services. |
| Instrument | Research analytical summaries (RAS). |

Following the above, the phases that were carried out for the analysis of the information and subsequent results are described in Figure 2.

**Phase 1. Definition of the subject of analysis**

Definition of the field of analysis of innovation and sustainability by establishing the following key words: sustainability, innovation, eco-innovation, sustainable innovation, processes, products.

**Phase 2. Literature review**

Bibliographic review of the SCOPUS database using TITLE - ABS- KEY (sustainable innovation" AND processes AND goods AND services AND products AND industries AND enterprises) finding a total of 62 articles.

**Phase 3. Analysis of the literature review**

Classification of the articles according to criteria by year, orientation and subject matter and analysis of the relationships between variables using Bibliometrix software.

**Phase 4. Discussion and results**

Contemplate from the analysis, the different scenarios of sustainable innovation from the processes, products and services and identify the aspects of discussion.

**Phase 5.Conclusions**

To conclusively estimate how sustainable innovation has impacted processes, products and services.

**Figure 2.** Description of the methodology and its phases.

### 3. Results

Worldwide, implementation of sustainability has been generated in several fields, such as in the production of materials, where according to [16], several methods of utilization of metal structures and reduction in carbon dioxide in primary production, recycling, and design of such facilities or metal structures are mentioned. Considering the rapid growth in the metal mechanics sector, which according to the referenced authors is expected to grow by 200% by 2050, we take into account that the useful life of metals is affected by physicochemical processes that occur between the exposure of the metal and environmental meteorological factors.

However, when reviewing the percentage of coverage of urban or rural areas based on sustainable processes, products, or companies, it is important to highlight what was stated by [17], who states that innovation cannot achieve its goal without adequate cash flow. In this sense, it is necessary to indicate that investment in the environmental component becomes a catalyst for sustainable innovation.

This is how, for example, when comparing the country of Ukraine with the United States, the former ranks 27th in the world ranking of countries with the lowest levels of green urban zones per person. In continuing to explore the trend of sustainability in the country of Ukraine, it is important to highlight [17], who state that corporate sustainability can be based on four elements: analysis of best practices and global trends, determination of the impact of companies on the social and environmental status, evaluation of the quality of the relevant certification(s) and analysis of compliance with social indicators of sustainable development [18].

When searching with the equations, the following information was found in the SCOPUS database using the following search term:

"sustainable innovation" AND "processes" AND "goods" AND "services" AND "products" AND "industries" AND "enterprises" AND PUBYEAR > 2018 AND ( LIMIT-TO ( OA,"all" ) ) AND ( LIMIT-TO ( PUBSTAGE,"final" ) ) AND ( LIMIT-TO ( DOCTYPE, "ar" ) ) AND ( LIMIT-TO ( PUBYEAR,2019) OR LIMIT-TO ( PUBYEAR,2020) OR LIMIT-TO ( PUBYEAR,2021) OR LIMIT-TO ( PUBYEAR,2022) OR LIMIT-TO ( PUBYEAR,2023) ) AND ( LIMIT-TO ( EXACTKEYWORD,"Sustainable" ) OR LIMIT-TO ( EXACTKEYWORD,"Sustainable Development" ) OR LIMIT-TO ( EXACTKEYWORD,"Innovation" ) ).

A total of 62 articles related to the search need were obtained. It is important to highlight that it was chosen to select the year 2018 to generate a more recent search for innovation models implemented taking into account the technology present in the sector and the degree of implementation of the Sustainable Development Goals SDGs formulated by the United Nations. Open-access articles were also included as a limiting variable to analyze the objectives, methodology, results, and conclusions of each study.

Concerning the analysis, it is important to relate the application of the Bibliometrix software, where the different variables such as dataset, sources, and authors, among others, were analyzed.

Regarding the countries analyzed that have implemented sustainable innovation strategies or models, in 10 articles, sustainable innovation in the United Kingdom is mentioned, while 9 papers relate to Sweden. As for the year of publication, it stands out that in the year 2022, 16 articles were published, while in the year 2021, there was a total of 15 articles. Finally, in 2020, there were 11 publications and in 2019, 10 research projects related to sustainable innovation were presented, as shown in Table 2.

**Table 2.** Published articles related to the topic by year.

| Year | Published Articles |
|------|--------------------|
| 2023 | 10 |
| 2022 | 16 |
| 2021 | 15 |
| 2020 | 11 |
| 2019 | 10 |
| **TOTAL** | **62** |

Precisely, it is important to highlight that the highest peak in research related to sustainability is presented in 2022, which may be a result of the worldwide economic activation in many companies from the culmination of the pandemic by COVID-19, so many organizations contemplated the inclusion of sustainable innovation as a strategy to improve competitiveness the opening of new markets.

Regarding the orientation towards the area of sustainability, it is evident that 47% of the research is focused on strengthening processes, while 51% of the articles studied indicate that the business sector is focused on the development of products with sustainable innovation criteria. Then, the production and articulation of service provision oriented towards sustainable innovation are the main focuses on which the company and industry orient their efforts to incorporate sustainability into their strategy.

Figure 3 shows which countries have published the most results in terms of research on the application of sustainable innovation at the business, product, and service levels. Sweden leads in the generation of knowledge through the publication of research articles, while Italy and the United Kingdom present nine publications.

It is important to highlight the behavior of publications worldwide, since, for example, in Europe, there has been a significant increase in publications related to sustainable innovation, while in South America, there have only been a few publications in Colombia and Brazil, which is precisely the result of the two most important conferences on sustainable development—firstly, the United Nations Conference on Environment and Development( Rio 92) and secondly, the United Nations Conference on Sustainable Development( Rio + 20).

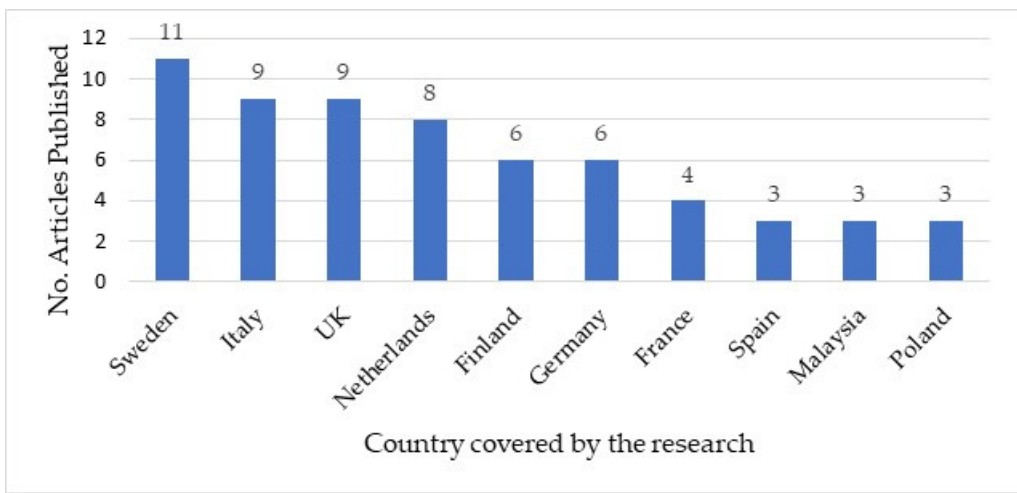

**Figure 3.** Countries participating in the research reviewed.

Now, according to Figure 4, the countries highlighted with a blue dark color are those that have led the implementation of strategies related to sustainable innovation, as is the case of Sweden, the United Kingdom, and Italy.

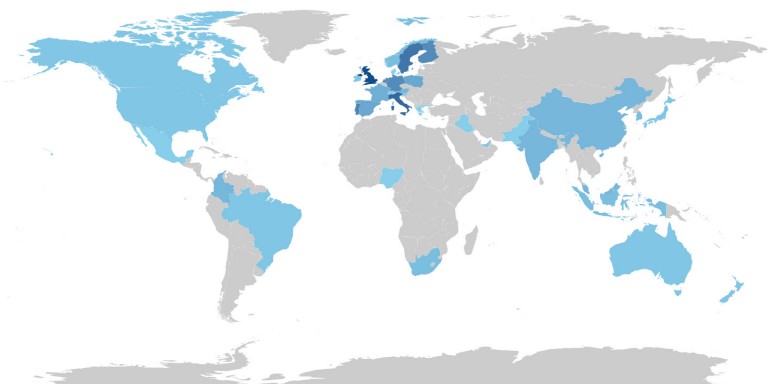

**Figure 4.** Scientific production by country.

Regarding the number of authors who analyzed the implementation of sustainable innovation in organizations, it is worth highlighting that there was a participation of 206 authors in the 62 scientific articles, 30 sources of information, and 34 average citations per paper. In total, 38.7% of the articles present the participation of foreign co-authors, which promotes the inclusion of points of view or perceptions regarding the implementation of sustainable innovation in their regions.

Concerning the words most used in each published article, and which are related to the object of the research, it is important to highlight that "sustainable development" has the highest percentage of documentary repetition, followed by sustainability, and finally the word "innovation", according to the word cloud provided by the Bibliometrix software.

According to Figure 5, the word with the most occurrences is "sustainable development" with 29 repetitions in most of the articles analyzed, and in second place is "sustainability". Now, it is worth highlighting that, for example, "circular economy" only had four occurrences, which indicates that, although it is true, the circular economy is a strategy for sustainable development (Almeida & Díaz [19]). It is not the only way to achieve ecological and economic benefits and promote sustainable consumer behavior, achieving greater employment opportunities [19].

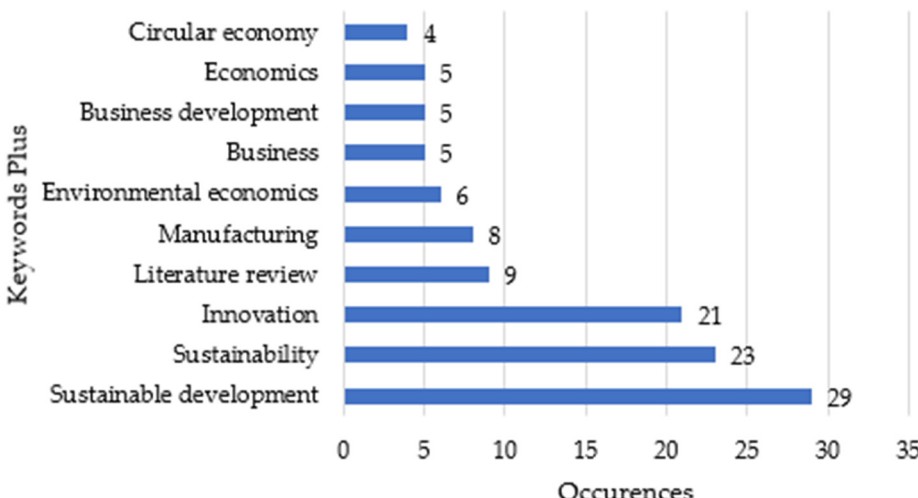

**Figure 5.** Frequently used words.

Now, on reviewing the co-occurrence in the research carried out, it is important to point out that according to Bibliometrics, the words located on the periphery of the network constitute the basis of the formulation of the different investigations, so, for example, it is emphasized that industrialization is the origin for conceiving changes in the manufacture of products or elements and this undoubtedly leads to innovation.

According to the factor analysis calculated through the Bibliometrix application, there is a strong correlation between innovation, production systems, manufacturing, and adaptive management to change. Now, it is necessary to emphasize that business innovation models are present in several studies, thus possessing a dark red color; however, their degree of closeness or affinity with production systems seems not to be very closely linked, i.e., business models do not necessarily lead to a change in production systems. It is also necessary to show or present the situation of supply chain management since it is quite close to business economics and automation, which represents a positive effect for those organizations making a change in their supply chains.

It is important to consider that depending on niche topics, most research focuses on the profitability that can be generated by implementing businesses where the circular economy and sustainable value are contemplated. Now, concerning the topics that become the driving force of the articles addressed, there are sustainable development, policy formulation, innovation in business models, and the relationship between stakeholders, aspect that can be seen in Figure 6.

Regarding the topics worked on, the work of [20] stands out, which relates to the results of an investigation of the implementation of circular business models in five industries, where high-value manufacturing (HVM in English) was contemplated as an inclusion variable.

In this sense, this research focused on analyzing and evaluating the relationship between the theory of circular business models (CBMs) with the current adoption of CBMs and their implementation in high-value manufacturing. Thus, the research addressed variables such as value, cost, and influence factors, which are elements that affect organizations seeking to implement the circular economy in their companies.

For the case of Indonesia, we highlight the work published by [21], who, through a review of the scientific literature on sustainable innovation in developing countries, analyzed the role of a sustainability leader in achieving sustainable innovation in the organization and, ultimately, long-term business success. It is important to highlight that the research generated a conclusion, among others, that sustainable innovation is the key for small and medium-sized companies to remain competitive.

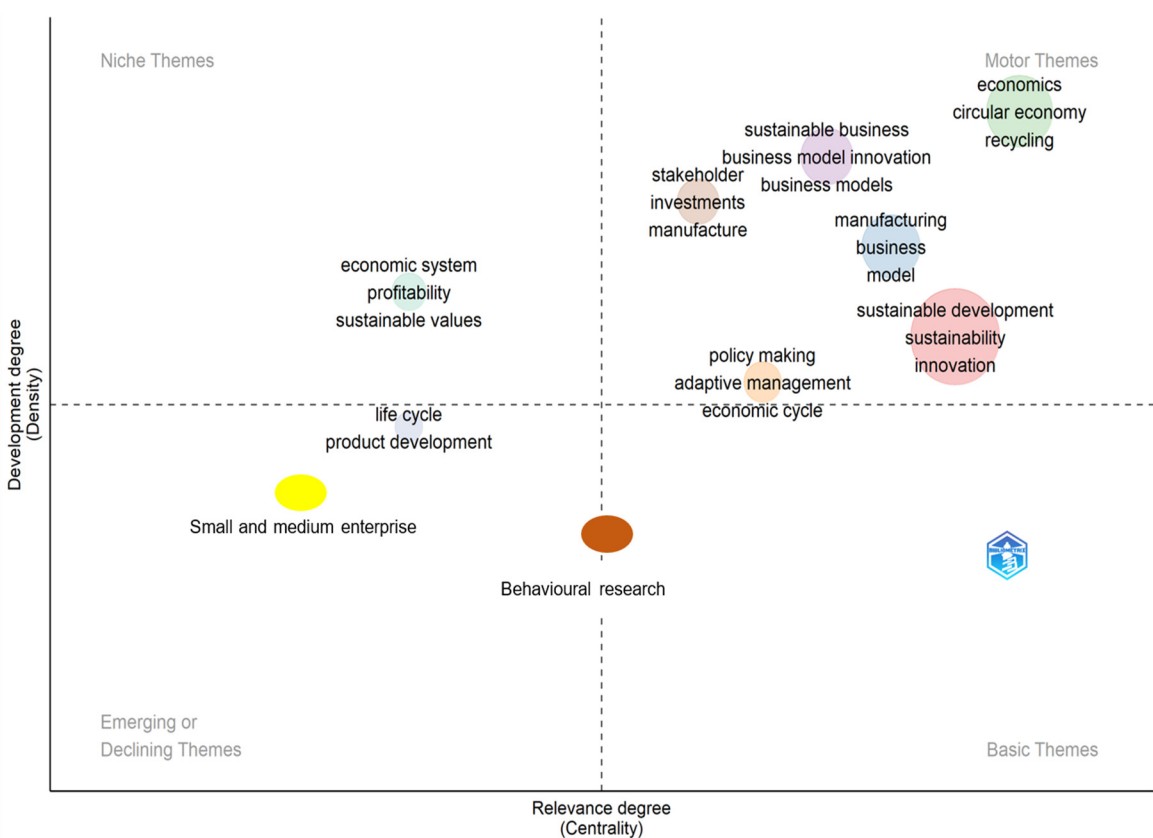

**Figure 6.** Factor analysis. Source: Bibliometrix.

For example, when analyzing the food sector, according to the study, such activity depends on the innovation potential of employees and the innovation culture of an organization, these variables being influenced by the support of sustainable innovation leaders. Figure 7 illustrates the characteristics that employees should have when implementing a sustainable innovation model.

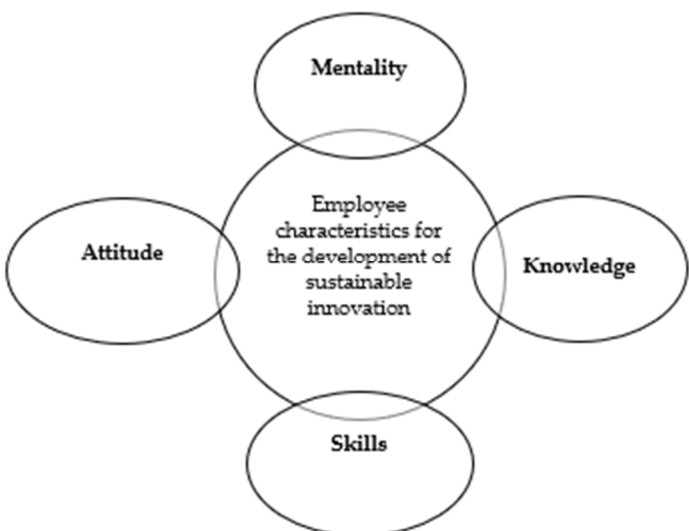

**Figure 7.** Employee characteristics for the development of Sustainable Innovation. Source: [21].

At the engineering level, the research work of [22] presents a systematic method called the waste bank, where samples of electronic waste are identified to determine the composition of the waste, and an electronic management model of the waste bank is

implemented as a sustainable circular economy. At this point, it is important that the government makes national regulations on e-waste and develops cooperation between local governments, industries, recycling producers and associations, and the community to build an e-waste collection system.

Likewise, it is important to highlight the study by [23], where network strategies for eco-innovation or sustainable innovation in manufacturing are explored from the point of view of industrial, academic, and governmental experts, to what is called the "Triple Helix", i.e., to implement sustainable innovation according to the above research, it is important to have the triangulation of the three perspectives.

The previous study focused on the analysis of eight topics among which the following are mentioned:

1. Commitment to stakeholder engagement.
2. Corporate assessment support.
3. Capacity building.
4. Protection of information.
5. Preparation for the transition.
6. Awareness campaigns.
7. Management audits.
8. Plant designs.

The research refers to intervention in three critical factors to potentiate eco-innovation which are as follows: networking platforms and forums, strategic alignment, and roadmaps. It is important to highlight that the definition of roadmaps refers to the definition of policies and strategies to promote the implementation of sustainable innovation.

Likewise, it is highlighted that nowadays, modern companies can use changes in their environment as opportunities to create products and services and thus achieve a competitive advantage [24].

## 4. Discussion

The alignment of strategies with sustainable development poses a relevant agenda in the business sector. Innovation, infrastructure, and the search for new technologies to mitigate the causes of environmental problems bring with them a new way of facing the challenges that, from the point of view of industry and business, are necessary to stop being a theory and be more internalized to the strategic plans and become reality, translated into investment, research and knowledge transfer.

The exploration of different arguments that consider sustainability as a central axis in the development of strategies for social, environmental, and economic involvement, directly influences the planning and determination of companies and industries to incorporate within their processes, products, and services, criteria for sustainable innovation. It is no longer only thought from the financial context as it was a few years ago, but a new model is being configured, more integrated, and accessible to all stakeholders.

The concept of sustainable innovation, although it does not have a structured technical definition at present, is understood as a set of strategies that seeks to incorporate environmental, social, and economic variables using tools such as information technology to increase productivity and improve the financial results of organizations. Some authors argue that sustainable innovation is a consistent model that allows not only the processes to benefit from having triple-bind criteria, but also, in the manufacture of products and provision of services, it is also perceived as something intrinsic that positively characterizes its use.

Therefore, in more developed countries, more research has been conducted on sustainable innovation, understanding that it is not an individual dimension, but rather one that converges several actors such as the collaborators themselves, investors, academia, and the governmental sector, among others. These investigations show that not only processes should incorporate these criteria, but also various services and products.

To this end, all the variables that can indeed lead to a sound strategy and sustainability over time must be considered at the outset, but also those factors that can hinder the implementation of these initiatives. Ref. [25] identifies which barriers and drivers exist in an innovation process. In this sense, to carry out a detailed analysis of the 62 articles, the inclusion of the following categories was reviewed to review the approach of each article to sustainable innovation at the organizational level. The Categories related to sustainable innovation of published papers is described in Table 3.

**Table 3.** Categories related to sustainable innovation of published papers.

| Intervention Area | No. References in Published Articles |
| :---: | :---: |
| Open innovation–innovation | 12 |
| Sustainable business models | 61 |
| Circular business model of innovation (CMBI) | 43 |
| Sustainability food sustainability | 31 |
| Industry 4.0 | 19 |
| Green quality circle | 4 |
| Corporate social responsibility | 27 |

According to the above, 44% of the published articles are related to open innovation, 26% are research focused on the inclusion of sustainable business models, while nine articles correspond to 9% whose topic is the circular business innovation model (CMBI), as shown in Figure 8.

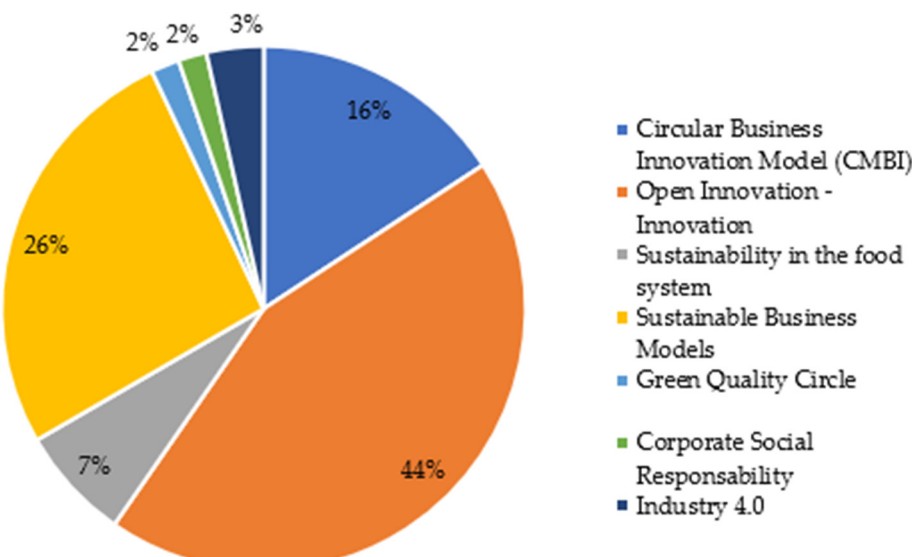

**Figure 8.** Distribution of the categories related to sustainable innovation in the published articles.

In this sense, it is important to highlight that, of the 62 articles found, 25 are related to open innovation or simply the implementation of innovation in the improvement of products or services that in a transversal way helps to reduce environmental impacts. Meanwhile, 15 studies emphasize business models that consider sustainability as their central axis, while 9 are related to sustainable innovation generated from the circular economy or circular business models. For example, Ref. [26] highlights that Japanese companies are more innovative, more competitive, and achieve better performance, which has resulted from the implementation of business practices such as culture management, open innovation, and innovation measurement, among others.

Now, regarding sustainable business, it is important to highlight tools or strategies that can be generated between the business context and the consuming public, as highlighted by [27], who mentioned 34 types of connections between companies, households, and the

state to enable economic gains and climate change mitigation, among which the "win–win" strategy where not only the economy of the organizations is analyzed but also a positive impact on the consumer, can be highlighted.

However, it is important to highlight that although it is true that it is possible for organizations to generate products based on eco-innovation, it is important to strengthen the marketing of these among consumers; for example, in countries such as Slovakia, authors such as [3] proposed a learning algorithm to positively influence among buyers and thus diminish the problem of the double externality that arises between the public and private benefits that are inherent to sustainable innovation or eco-innovation.

Likewise, it is not viable to speak of sustainable business if there is no creation of state policies that help companies in their commitment to climate change and that allow for the strengthening of different actors in the supply chain of products [28].

Concerning the food sector, it is highlighted that sustainable innovation has permeated meat production, mentioning the generation of in vitro meat by the culture of animal stem cells to achieve a sustainable global food system [29]. It not only focuses on the production of this type of food but also on the use of renewable energies in the food sector to achieve the Sustainable Development Goals (SDGs) [30].

Faced with the fourth industrial revolution, or Industry 4.0, it is important to highlight authors such as [31], who refer to additive manufacturing (which is associated with 3D printing) as a new model in industrial sustainable business, provided that a transformation in the industry and acceleration in the rate of adoption of new technologies can be achieved to achieve a decrease in social, environmental, and economic impacts.

In the Table 4, it is possible to identify such variables that can put at risk the implementation of an innovation model, but in turn which are those criteria that can positively impact its implementation.

**Table 4.** Variables considered obstacles and drivers that can be applied to sustainable innovation. Source: Adapted from [25].

| Barriers | Thrusters |
|---|---|
| Bureaucracy | Knowledge organizational knowledge |
| Difficulty technical difficulty | Understanding of the needs of staff. |
| High costs for access to innovation | Support policies and protocols |
| Difficulty in banking | Training and coaching |
| Organizational conflicts | Concise and appropriate information. Knowledge networks |
| Uncertainty about the results | internal and external knowledge of the environment |
| Lack of information | |

Analyzing the axes or orientations of sustainability, 51% of the articles analyzed are focused on sustainable product innovation. Everyday **products** have incorporated sustainability practices, particularly in the raw materials with which they are made. Ref. [32] refers to the fact that the value chain of the life cycle of products is aligned with the needs of consumers and, therefore, products must be ready every day so that their useful life is longer and thus generate less pressure on natural capital.

On the other hand, when analyzing the orientations in sustainability, it should be noted that 47% of the research analyzed is more focused on **processes**; this implies that the business sector is integrating its different work methods with a more sustainable vision and above all is aligned with the agendas in this area, which strengthens them competitively. Ref. [33] argues that a sustainable business model is a key success factor for being competitive and has a direct connection to success in business.

The **services** and sustainable innovation whose studied items are equivalent to 2% are related from the strategic background. Ref. [34] maintains that the benefits of a sustainable

business model are immersed in the value proposition, an aspect that directly impacts service delivery.

Because of the above, it would be worthwhile to establish lines of action that lead organizations to increasingly adopt initiatives that are in line not only with their internal context but also with the external context, such as the adoption of best practices for manufacturing products or providing services. To be able to identify where there is an area of opportunity for growth, but also what can hinder it, is important to understand the true organizational capabilities to meet these challenges.

It is important to consider the concept of innovation established by the Oslo Manual [2], which indicates that this strategy may consist of an improved product or process that differs from the previous or old process. Thus, it is important to highlight that one way to transform a process in the sale of an element or input through innovation can be through the incursion of eco-innovation in marketing, in which, according to [3], if interest is generated in the acquisition of ecological products, a change in culture is encouraged in stakeholders and minimize negative or adverse aspects towards sustainable innovation.

Likewise, it is pertinent to mention the characteristic of novelty within innovation, considering that novelty can arise from new ideas, models, methods, or prototypes [2], so, in the case of the wine industry, by introducing photovoltaic roofs, lighter bottles, an improvement is generated both at the level of the process, as well as the product and a minimization of the environmental impact [35].

Regarding the existing types of innovation, according to [36], it can be said that all of the different scenarios presented could contribute in one way or another to strengthening initiatives in processes, products, and services. For example, the degree of transformation described there focuses on incremental, static, or radical changes that can occur according to the context presented and depending on the variables that qualify the type of product, the process environment, or the way of providing the service.

As far as the functional and operational areas are concerned, they depend to a large extent on the management of resources, the adequacy of human resources, and the use of technologies, which, together with an innovation strategy, will allow them to be sustainable over time. As far as the offer is concerned, it is important to fully identify to whom innovative products and services are being offered in order to identify the corresponding niche, since it is not possible to offer this type of service to interest groups that do not have compatibility with what they need. When talking about how to implement operational strategies, attention is focused on the type of activities to be carried out and on having integral collaborative teams that favor continuous processes that lead to sustainability.

## 5. Conclusions

Sustainable innovation began with important work in the manufacture of products, always focusing on reducing the adverse impacts that they cause or in effect introducing appropriate technologies to help combat damage to the environment where they had an effect. Subsequently, it was required to work in-depth on the strategy, and with it came the processes, which brought to the business sector order, method, and standardization of activities.

To bring sustainable innovation closer to organizations, it is necessary to link technology with systemic sustainability, which although it is not everything, is a great driver, so the integrality between existing and complementary processes can interact positively and thus generate better products.

Likewise, digital production and information technologies, for example, in the development of logistics systems, have economic, social, environmental, and other effects.

In a second instance, there are the **policies and governance** of sustainable innovation since guidelines are necessary in the implementation of such a model, so much so that governmental portfolios, speaking of ministries of science and technology, place innovation as the center of action for the articulation of knowledge under a local, national, and global criterion. Thus, "green" legal regulation has managed to influence decision-making in

the implementation of sustainability, where in some cases, companies have manipulated the creation of environmental policies to obtain advantages in their relationship with climate change.

Likewise, it is important to highlight as an **agent of change the normative and legislative** aspects that favor, for example, companies to access financing for projects with sustainable innovation components that directly impact society and economic growth, as stipulated in Colombia by Law 2294 of 2023 whereby the National Development Plan 2022–2026 "Colombia World Power of Life" is issued, and Law 1964 of 2019 whereby the use of electric vehicles in Colombia, among other legal regulations not only in Colombia but also in other countries such as the United Kingdom and Switzerland.

Another focus is the **people and networks of knowledge** as a product of the co-creation between the competence of the personnel involved in the projects as such of innovation and the education and training in this field. Last but not least, but no less important is **participation**, conceived from the exercise where the companies and the people who are part of it thus facilitate the process of building thinking in the process, sustainable and innovative products and services, where through the transfer of knowledge, large multinationals, as well as small and medium-sized companies, can incorporate long-term sustainability practices improving organizational performance.

**Author Contributions:** Conceptualization, J.Z., G.O., D.B. and D.G.; methodology, J.Z., G.O., D.B. and D.G.; software, Bibliometrix version 4.1.4.; validation, J.Z., G.O.; formal analysis: D.B. and D.G.; investigation, J.Z., G.O., D.B. and D.G.; resources, J.Z., G.O., D.B. and D.G.; data curation, J.Z., G.O., D.B. and D.G.; writing—original draft preparation, J.Z., G.O., D.B. and D.G.; writing—review and editing, D.B. and D.G.; visualization, J.Z., G.O., D.B. and D.G.; supervision, J.Z., G.O.; project administration, J.Z., G.O.; funding acquisition, J.Z., G.O. All authors have read and agreed to the published version of the manuscript.

**Funding:** This research received no external funding.

**Data Availability Statement:** The data on which this article is based are supported by different data portals such as national government pages, repositories, and other data sources.

**Conflicts of Interest:** The authors declare no conflicts of interest.

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
