# Peer review of "Sustainable Innovation in Organizations: A Look from Processes, Products, and Services"

_sustainability, doi:10.3390/su16062503_

Round 1

Reviewer 1 Report

Comments and Suggestions for Authors

The authors analyzed 62  manuscripts in the span 2018 to 2023. First of all, there should be more articles to provide a real literature review. 

The first 4 paragraphs from 2. Materials and Methods should be avoided. 

Sections Innovation and sustainable development and Companies and sustainability must be re-written.

Figures 3, 4, 7 and 8 should be avoided. Figures Figure 10.  and 11 are too blury.

Discussion and conclusion should be much better providing the novelty of the research, directions, analysis....

Comments on the Quality of English Language

Proof reading. 

Author Response

Dear Reviewer,

Thank you for your comments and contributions to our paper, below you will find the responses to your observations:

  1. The authors analyzed 62 manuscripts in the span 2018 to 2023. First of all, there should be more articles to provide a real literature review.

At the time of executing the search equation and including the variables to be analyzed, some extra articles were returned, but they did not really contribute to the analysis of the topic presented in the article. More recent articles have used a number close to the one presented.

  1. The first 4 paragraphs from 2 Materials and Methods should be avoided

This suggestion was heeded and corrected.

  1. Sections innovation and sustainable development and companies and sustainability must be re-written.

The corresponding text was adjusted to make it more understandable and comprehensible according to the subject matter.

  1. Figures 3, 4, 7 and 8 should be avoided. Figures 10 and 11 are too blury.

These figures provide a better understanding of the analysis being performed. As for the blurred figures, they have been eliminated and/or adjusted according to suggestions.

  1. Discussions and conclusion should be much better providing the novelty of the research, directions, analysis.

They have been adjusted according to suggestions

Reviewer 2 Report

Comments and Suggestions for Authors

1 It is not clear what the research problem is, so to speak. In the introduction, the authors write "Therefore, the problem arises of determining how organizations in the Latin American region manage sustainable innovation in their processes” [34-36]. The research section, on the other hand, examines the global body of literature.

2. The authors write: "The purpose of this article is to identify what are those factors that have allowed or not to implement processes, product development, and services within the context of sustainable innovation, which will allow us to solve the question: how has the process of implementing sustainable innovation at the organizational level in processes, products and services been? This will make it possible to analyze the different criteria that have been established at the organizational level to promote these practices, allowing different companies, regardless of size, to contribute to development and sustainability” [50-56]. However, the authors did not explicitly identify these factors - they resound to some extent at the end of the article, but it would be useful to summarize them explicitly.

3. There is no separate literature review section on sustainable innovation in the article. The literature section is presented in the "2. Materials and Methods" section. This is incorrect - the Literature review section should be separated from this section.

4. [190]: PUBYEAR >: will help to purge articles published after 2018 - it should be before 2018

5. [193]: Twice is sustainability

6. [496]: 2626% - I suppose it should be 26%

Author Response

Thank you for your comments and contributions to our paper, below you will find the responses to your observations:

  1. Is not clear what the research problem is, so to speak. In the introduction, the authors write "Therefore, the problem arises of determining how the organizations in the Latin American region is manage sustainable innovation in their processes" [34-36]. The research section, on the other hand, examines the global body of literature.

The scope of the research was adjusted considering that not only LATAM will be taken as a region to analyze the effects of the implementation of sustainable innovation, but that a global review was included, taking into account the results of projects related to the theme. Thus, the Latin American region was removed as a research area from the research study.

  1. The authors write “The purpose of this article is to identify what are those factors thar have allowed or nor implement processes, product development, and services within the context of sustainable innovation at the organizational level in processes, product and services been? This will make it possible to analyze the different criteria that have been established at the organizational level to promote these practices, allowing different companies, regardless of size, to contribute to development and sustainability” [50-56]. However, the authors did not explicitly identify these factors – they resound to some extent at the end of the article, but it would be useful to summarize them explicitly.

The approach that was carried out in the analysis by focus topics on the result of the implementation of sustainable innovation and on the business model adopted according to the commercial sectors, is indirectly related to the form or effect that the innovation has generated. innovation. sustainable in processes, products, and services. For example, when referring to the circular economy, this strategy implies an implementation of sustainable innovation in processes and products, since in most cases a reengineering is carried out in the processes to modify the characteristics of the products or elements and their life cycle. Likewise, in manufacturing, the implementation of sustainable innovation in processes and products can be analyzed with greater proportionality than in the service sector.

  1. There is no separate literature review section on sustainable innovation in the article. The literature section is presented in the “2 Materials and Methods” section. This is incorrect – The literature review section should be separated from this section.

It was adjusted in the document in the respective section.

  1. [190]: PUBYEAR >: will help to purge articles published after 2018 -it should be before 2018

Research prior to 2018 was not taken into account, considering that according to Hernández Sampieri, et al, (2016), it is recommended to take publications to carry out a literary review maintaining an interval of five (05) years.

  1. [193] Twice is sustainability

It was adjusted in the text

  1. [496]: 2626% - I suppose it should be 26%

It was adjusted in the text

Reviewer 3 Report

Comments and Suggestions for Authors

Dear authors,

Thank you for sending your paper to the journal; although the topic is interesting, the paper is in the early stage and needs further efforts in the following sections:

1-The originality of the paper is not fully justified

2-What are the research contributions?

3- The theoretical issues should be included in the paper

4-The analyses section should benefit from other tests to validate the primary results

5-The conclusion looks very general and should be revised according to the findings

6-The practical and managerial implications should be stated based on the results.

Comments on the Quality of English Language

Needs to conduct proofreading

Author Response

Dear Reviewer, 

Thank you for your comments, adjustments and contributions to our paper, below you will find the responses to your observations:

  1. The paper is in the early stage and needs further efforts in the following sections

It has been adjusted in accordance with the suggestions of all reviewers. Further amplification has been included in the different paragraphs of the article.

  1. The originally of the paper is not justified

The document has been adjusted in its justification as well as in the bibliography supported with complementary analyses according to the request made by the reviewer.

  1. What are research contributions

The contributions to this work are based on the systematic review of the data provided by previous studies that show the behavior of innovation in sustainability and the trend in this field for products, services, and processes.

  1. The theorical issues should be included in the paper

The theoretical bases are supported by the 62 bibliographic references consulted for this article, which are supported and analyzed. The analyses section should benefit from other tests to validate the primary results

  1. The analyses section should benefit from other tests to validate the primary results

Being a systematic review, validations were considered to draw the conclusions of the article

  1. The conclusions looks very general and should be revised according to the findings

Other conclusions were added that strengthen the document.

  1. The practical and managerial implications should be stated based on the results

In the analysis carried out, the implications that sustainable innovation brings with it for the variables analyzed can be seen. These conclusions were enriched with other contributions made.

  1. Needs to conduct proofreading

Comments were reviewed and supported within the article, as well as in this document.

Best regards,

Round 2

Reviewer 1 Report

Comments and Suggestions for Authors

Authors followed recommendations. 

Comments on the Quality of English Language

Proof reading. 

Author Response

Dear Reviewer,

Thank you for all your recommendations and suggested adjustments, we now have an improved paper, including a new revision of the English.

Best regards,

Reviewer 3 Report

Comments and Suggestions for Authors

Dear authors,

Thank you for sending your revised paper; the current version meets my academic expectations.

Author Response

Dear Reviewer,

Thank you for all your recommendations, we now have an improved paper, including a new revision of the English.

Best regards,